# Mechanistic study on the side arm effect in a palladium/Xu-Phos-catalyzed enantioselective alkoxyalkenylation of γ-hydroxyalkenes

Shuai Zhu[1,5], Zihao Ye[2,5], Ming-Jie Chen[1,5], Lei Wang[3], Yu-Zhuo Wang[1], Ke-Nan Zhang[1], Wen-Bo Li[1], Han-Ming Ding[1], Zhiming Li ®[2] ✉ & Junliang Zhang ®[1,2,3,4] ✉

Recently, the asymmetric bifunctionalization of alkenes has received much attention. However, the development of enantioselective alkoxyalkenylation has posed a considerable challenge and has lagged largely behind. Herein, we report a new palladium-catalyzed enantioselective alkoxyalkenylation reaction, using a range of primary, secondary, and tertiary γ-hydroxy-alkenes with alkenyl halides. By employing newly identified Xu-Phos (**Xu8** and **Xu9**) with a suitable side-arm adjacent to the PCy$_2$ motif, a series of allyl-substituted tetrahydrofurans were obtained in good yields with up to 95% *ee*. Besides (*E*)-alkenyl halides, (*Z*)-alkenyl halide was also examined and provided the corresponding (*Z*)-product as a single diastereomer, supporting a stereospecific oxidative addition and reductive elimination step. Moreover, deuterium labeling and VCD experiments were employed to determine a *cis*-oxypalladation mechanism. DFT calculations helped us gain deeper insight into the side-arm effect on the chiral ligand. Finally, the practicability of this method is further demonstrated through a gram-scale synthesis and versatile transformations of the products.

Tetrahydrofuran derivatives bearing substituents at the C2 position are important subunits in pharmaceuticals, materials, and natural products (Fig. 1)[1–8]. Thus, many methods for stereoselective synthesis of multisubstituted tetrahydrofurans have been developed[9–24], among which metal-catalyzed cyclizations of readily available unsaturated alcohols have shown good performance in the synthesis of diverse tetrahydrofuran products with predictable regio- and diastereo-selectivity[25–39]. The groups of Zurek, Chemler[40,41], and Liu[42] independently developed copper-catalyzed asymmetric radical alkene carboalkoxylation of γ-hydroxyalkenes, which provide facile access to chiral tetrahydrofurans with high enantioselectivities. Sigman and co-workers reported an efficient Pd/Cu co-catalyzed enantioselective alkene bifunctionalization reaction of γ-hydroxyalkenes via intramolecular oxypalladation and subsequent addition[43,44]. Using chiral salen-Co complexes, Kang achieved an elegant instance of enantioselective intramolecular iodoetherification to furnish such chiral five-membered oxygen heterocycles[45]. Regarding the significance of chiral tetrahydrofuran, the development of highly enantioselective

[1]Shanghai Key Laboratory of Green Chemistry and Chemical Processes, School of Chemistry and Molecular Engineering, East China Normal University, 3663 N. Zhongshan Road, Shanghai 200062, P.R. China. [2]Department of Chemistry, Fudan University, 2005 Songhu Road, Shanghai 200438, P.R. China. [3]School of Chemistry & Chemical Engineering, Yangzhou University, Yangzhou 225002, P.R. China. [4]School of Chemistry & Chemical Engineering, Henan Normal University, Xinxiang 453007 Henan, P.R. China. [5]These authors contributed equally: Shuai Zhu, Zihao Ye, Ming-Jie Chen. ✉e-mail: zmli@fudan.edu.cn; junliangzhang@fudan.edu.cn

methods for their synthesis from readily available starting materials, especially under mild conditions is still highly desirable.

Because alkenyl group can be converted into various functional groups, which is particularly important for increasing the diversity of compounds[46]. Several progresses have been achieved in alkenylation reactions, which provided efficient access to compounds with alkenyl framework. For instance, Sigman realized the aryl-alkenylation of unactivated alkenes with vinyl triflates and boronic acids furnishing skipped dienes, however, the substrates were limited to aryl 1,3-dienes[47]. Recently, Shu reported a cross-electrophile reaction of unactivated alkenes with vinyl triflates, which led to highly enantioselective aryl-alkenylation products including dihydrobenzofurans, indolines, and indanes in good yields with up to >99% ee[48]. Unfortunately, the related synthesis of enantiopure tetrahydrofurans with 2-allylic substituent from unactivated alkenols is rarely explored.

Wolfe and Tang's works have constructed a series of asymmetric oxygen-containing heterocyclic by Pd-catalyzed carboetherification reactions of alkenes with aryl bromides (Fig. 2a). However, the involvement of alkenyl bromides in the reaction has not been reported successfully, although Wolfe and Tang have made related attempts. The result indicates that the reactivity and enantioselectivity control of alkenyl bromine and aryl bromine in the reaction are very different, even though they are both C(sp²)-Br in form. Inspired by the good performance of sulfinamide-phosphine (**Sadphos**) in asymmetric catalysis[49–58] and the unsuccess of the alkoxyalkenylation in Wolfe and Tang's previous work[59–61], we became very interested in the Pd-catalyzed asymmetric alkoxyalkenylation reaction of γ-hydroxy alkenes with alkenyl halides, which remains as an unsolved problem and pose considerable challenges: 1) the catalyst inhibition caused by competitive side reactions, including Heck reaction, β-hydride elimination and double bond migration of the γ-hydroxyalkene;[31,59–61] 2) whether metal-catalyzed terminal olefin nucleopalladation is *cis* or *trans*-attack has always been an open question[49,50]; 3) the asymmetric synthesis of tetrahydrofurans with a tertiary carbon center in one-step, especially under mild conditions[41,42]; 4) primary, secondary and tertiary γ-hydroxyalkenes provide products in good yield and ee[59–61].

Herein, we report new sulfinamide phosphine ligands (**Xu8** and **Xu9**) with a side-arm adjacent to the PCy₂ motif, which are crucial to address the challenging enantioselective alkoxyalkenylation reaction of γ-hydroxy-alkenes with alkenyl halides under mild conditions (Fig. 2b). Moreover, VCD method and DFT calculation are utilized to verify the *cis*-oxypalladation mechanism, which is usually ambiguous in similar reactions. Transition states for the *cis*-oxypalladation step are located and analyzed, revealing the origin of the side-arm effect in our ligands.

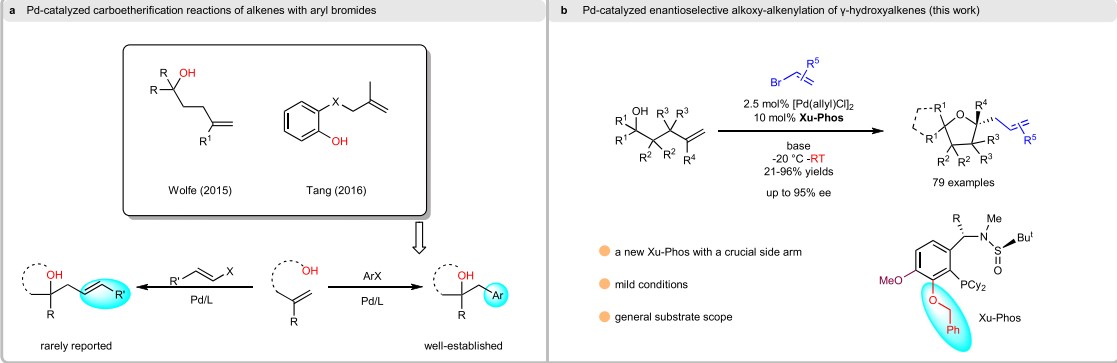

**Fig. 1 | Biologically active compounds containing 2-substituted tetrahydrofuran skeletons.** Substructures colored in blue highlight the core 2-substituted tetrahydrofuran skeleton with stereocenter. Zidovudine, an antimetabolite and a HIV-1 reverse transcriptase inhibitor. ALS⁻⁸¹¹², respiratory syncytial virus (RSV) polymerase inhibitor. Magnofargesin, a natural product found in *Magnolia biondii*. Fijianolide A, a natural compound isolated from *Spongia myco-fijiensis*. Ophiobolin A, a fungal secondary metabolite has activity against glioblastoma. Amphidinolide X, a natural product isolated from *Amphidinium sp*.

**Fig. 2 | Background and reaction design. a** Pd-catalyzed carboetherification reactions of alkenes with aryl bromides. **b** Pd-catalyzed enantioselective alkoxy-alkenylation of γ-hydroxyalkenes.

# Results

## Reaction Development

In our initial study, tertiary γ-hydroxyalkene **1a** and alkenyl bromide **2a** were selected as the model substrates. A series of privileged chiral ligands **L1-L8** were investigated, such as (*R*)-Tol-BINAP (**L1**), (*R*)-QuinoxP (**L2**), (*S,S*)-Me-duphos (**L3**), ferrocene ligands (**L4-L6**), (*S,S*)-DIOP (**L7**), (*R*)-BIDIME (**L8**), in which the (*R*)-BIDIME **L8** showed the promising result to deliver **3a** in 58% yield with 61% *ee* (Fig. 3). Next, we turned our attention to our developed sulfinamide-phosphine (**Sadphos**) ligand such as Xu-Phos (**Xu1**) with free N-H bond, unfortunately, it gives 33% yield and 2% *ee* (Fig. 3). To our delight, the N-Me-Xu-Phos (**Xu2**) could deliver the desired product in 81% yield with 85% *ee*. Then, we focused on adjusting the R group in the Xu-Phos (**Xu3, Xu4**), but no better results were obtained.

Inspired by Tang's findings that introducing a side-arm in the ligand is an efficient strategy to improve enantioselectivity by adjusting the electronic properties and space environment of the catalyst, like SaBOX ligands[62,63], we next designed and prepared new ligands **Xu5-Xu8**. These ligands bear a sterically congested substituent at the *ortho* position of the phosphine group (Fig. 3). The absolute configuration of **Xu6** was well established by X-ray crystallography analysis. Gratifyingly, ligand **Xu8**, easily prepared from 2-bromo-3-hydroxy-4-methoxybenz-aldehyde, delivers much better *ee* (92% *ee*) compared to **Xu6** (82% *ee*) under the conditions with the 2.5 mol% of [Pd(allyl)Cl]$_2$ and 10 mol% ligand, NaO$^t$Bu as the base, H$_2$O as the additive in Et$_2$O/hexane at RT. The introduction of a methoxy group into the *meta*-position of the phosphine group in ligand **Xu8** was critical to the stereoselectivity and reactivity. By changing *p*-Ph-C$_6$H$_4$ to 3,5-CF$_3$-C$_6$H$_3$, **Xu9** afforded the product in 63% yield with 90% *ee*. While changing OMe to OBn or OEt, **Xu10**, and **Xu12** gave comparative *ee* (89% *ee*, 90%

*ee*) compared to **Xu8**. Furthermore, changing OMe to O$^i$Pr (**Xu13**), led to a lower ee (85% *ee*) compared to **Xu8**. Compared to **Xu3, Xu11**, which contained a methoxy group in the *para*- and *meta*-position of the ligand **Xu3**, yielded slightly lower results (52% yield, 84% *ee*). We proposed that the methoxy group substituent pushes the side-arm OBn much closer to the catalytic center, and steric effects between methoxy or other alkyloxy (OR) and OBn groups may be crucial for the high stereoselectivity of Xu-Phos.

To obtain better enantioselectivity (vide infra), further screening of the reaction conditions was examined. As shown in Table 1, the absence of H$_2$O resulted in **3a** with slightly inferior enantioselectivity (Table 1, entries 2–6). Investigation of palladium catalysts revealed that [Pd(allyl)Cl]$_2$ was most effective in this transformation (Table 1, entries 7–9). Changes of bases to KO$^t$Bu, LiO$^t$Bu, and Cs$_2$CO$_3$ led to a significant decrease in enantioselectivity (Table 1, entries 10–13). Using the NaOH instead of NaO$^t$Bu, can improve the enantioselectivity to 97% *ee*, but decrease the yield to 29% (Table 1, entry 13). And further solvent screening failed to improve the enantioselectivity (Table 1, entries 14–17).

## Substrate scope of the reaction

With the optimal reaction conditions in hand (Table 1, entry 1), we evaluated the scope of alkenyl bromides (Fig. 4). The (*E*)-styrenyl bromide and phenyl substituents with electron-donating groups (Me, MeO) at *ortho*-, *meta*- and *para*-position worked well to form the corresponding compounds **3a-3e** in 51–74% yields with 90–92% *ee*s. Disubstituted and trisubstituted phenyl rings with OMe at different positions were also compatible, delivering the corresponding products **3f-3g** in 42–63% yields with 84–92% *ee*s. On the other hand, substrates with electron-withdrawing groups at *meta*- and

**Fig. 3 | Screened chiral ligands.** Reaction conditions: **1a** (0.1 mmol), **2a** (0.2 mmol), NaO$^t$Bu (4 equiv.), H$_2$O (2 equiv.), [Pd(allyl)Cl]$_2$ (2.5 mol%), **Xu8** (10 mol%) in 1 mL Et$_2$O/Hexane under N$_2$ at RT for 72 h. Isolated yields. *ee* was determined by chiral HPLC analysis.

**Table 1 | Optimization of the reaction conditions[a]**

| Entry | Additive | [Pd] | Base | Solvent | Yield% | Ee% |
|---|---|---|---|---|---|---|
| 1 | H₂O | [Pd(allyl)Cl]₂ | NaOtBu | Et₂O/Hexane = 1:1 | 74 | 92 |
| 2 | -- | [Pd(allyl)Cl]₂ | NaOtBu | Et₂O/Hexane = 1:1 | 84 | 84 |
| 3 | 4 Å MS | [Pd(allyl)Cl]₂ | NaOtBu | Et₂O/Hexane = 1:1 | 79 | 91 |
| 4 | MeOH | [Pd(allyl)Cl]₂ | NaOtBu | Et₂O/Hexane = 1:1 | 75 | 79 |
| 5 | EtOH | [Pd(allyl)Cl]₂ | NaOtBu | Et₂O/Hexane = 1:1 | 54 | 70 |
| 6 | iPrOH | [Pd(allyl)Cl]₂ | NaOtBu | Et₂O/Hexane = 1:1 | 52 | 60 |
| 7 | H₂O | Pd(dba)₂ | NaOtBu | Et₂O/Hexane = 1:1 | 43 | 93 |
| 8 | H₂O | Pd(OAc)₂ | NaOtBu | Et₂O/Hexane = 1:1 | 72 | 92 |
| 9 | H₂O | PdCl₂ | NaOtBu | Et₂O/Hexane = 1:1 | 12 | 78 |
| 10 | H₂O | [Pd(allyl)Cl]₂ | KOtBu | Et₂O/Hexane = 1:1 | -- | -- |
| 11 | H₂O | [Pd(allyl)Cl]₂ | LiOtBu | Et₂O/Hexane = 1:1 | 41 | 7 |
| 12 | H₂O | [Pd(allyl)Cl]₂ | Cs₂CO₃ | Et₂O/Hexane = 1:1 | 15 | 85 |
| 13 | H₂O | [Pd(allyl)Cl]₂ | NaOH | Et₂O/Hexane = 1:1 | 29 | 97 |
| 14 | H₂O | [Pd(allyl)Cl]₂ | NaOtBu | Toluene | 90 | 91 |
| 15 | H₂O | [Pd(allyl)Cl]₂ | NaOtBu | THF | 51 | 90 |
| 16 | H₂O | [Pd(allyl)Cl]₂ | NaOtBu | Et₂O | 36 | 94 |
| 17 | H₂O | [Pd(allyl)Cl]₂ | NaOtBu | Hexane | 55 | 88 |

[a]Reaction conditions: **1a** (0.1 mmol), **2a** (0.2 mmol), NaOtBu (4 equiv.), H₂O (2 equiv.), [Pd(allyl)Cl]₂ (2.5 mol%), **Xu8** (10 mol%) in 1 mL Et₂O/Hexane under N₂ at RT for 72 h. Isolated yields. ee was determined by chiral HPLC analysis.

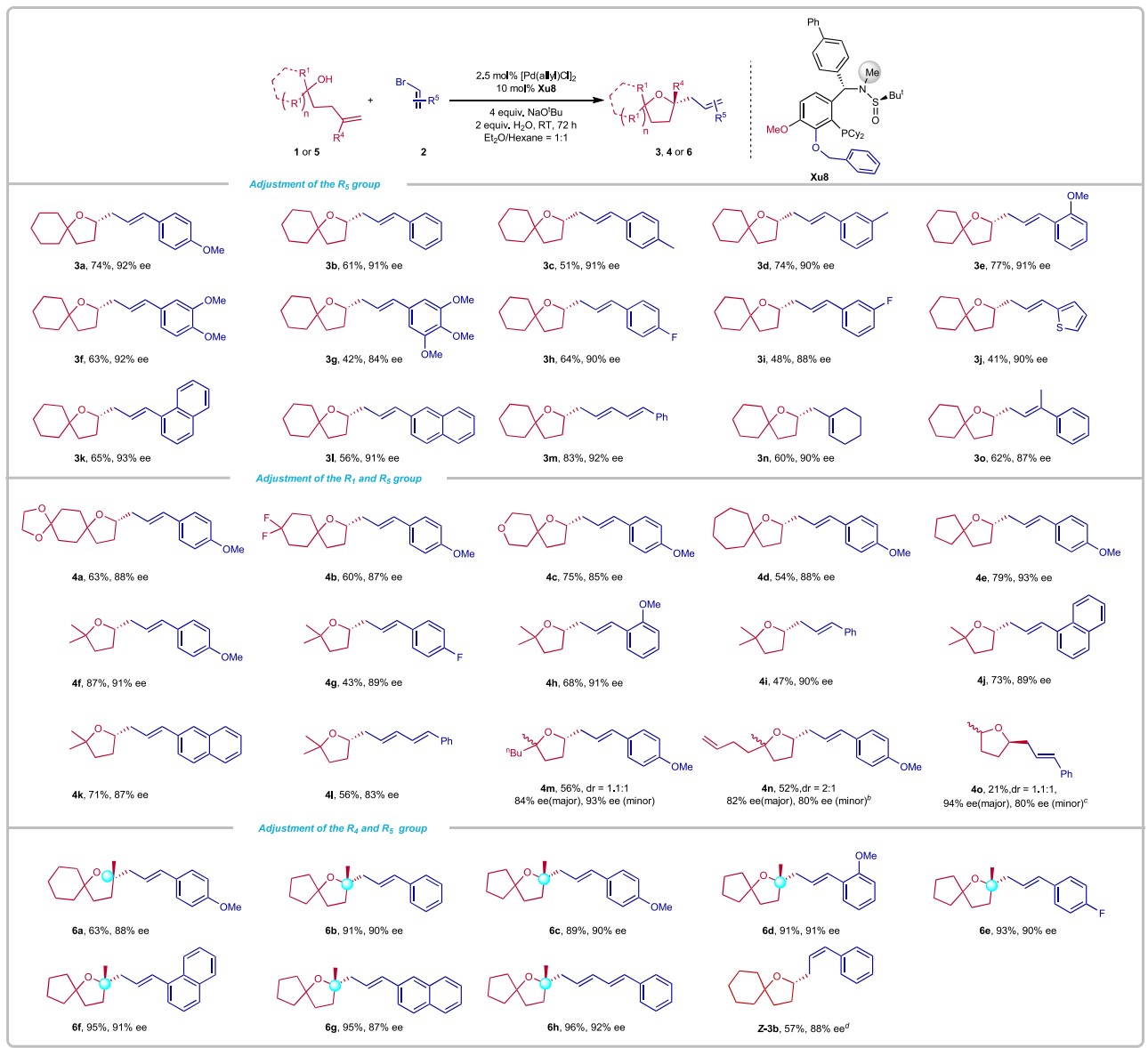

**Fig. 4 | Synthesis of 3, 4, or 6.** Reaction conditions: **1** or **5** (0.2 mmol), **2** (0.4 mmol), NaO$^t$Bu (4 equiv.), H$_2$O (2 equiv.), [Pd(allyl)Cl]$_2$ (2.5 mol%), **Xu8** (10 mol%) in 2 mL Et$_2$O/Hexane under N$_2$ at RT for 72 h. Isolated yields. *ee* was determined by chiral HPLC analysis. $^b$ with **Xu6** (10 mol%) in 2 mL toluene under N$_2$ at RT for 72 h. $^c$ **1** (0.5 mmol), 2 (1 mmol), EtONa (4 equiv.), [Pd(allyl)Cl]$_2$ (2.5 mol%), (R, R$_S$)-**Xu9** (10 mol%) in toluene under N$_2$ at −20 °C for 72 h. $^d$ 5 mol% [Pd(allyl)Cl]$_2$, 20 mol% **Xu2**.

*para*-position also proceeded well and generated **3h-3i** in 48–64% yields with 88–90% *ee*s. The phenyl group could be replaced by heteroaryl or other aryl groups, such as 2-thienyl, 2-naphthyl, and 1-naphthyl, delivering the desired **3j-3l** in 41–65% yields with 90–93% *ee*s. Furthermore, chain alkenyl bromide, cycloalkenyl bromide, and multi-substituted bromide were investigated, furnishing the relevant products **3 m, 3n**, and **3o** in 60–83% yields with 87–92% *ee*s. Next, we investigated different types of tertiary γ-hydroxyalkenes. We found that functionalized tertiary γ-hydroxyalkenes bearing the electron-donating (MeO) and electron-withdrawing (F) groups on the six-membered rings were well tolerated to afford the corresponding products **4a** and **4b** in 60–63% yields with 87–88% *ee*s. The pyran-derived tertiary γ-hydroxyalkene was also suitable for this transformation, delivering **4c** in 75% yield with 85% *ee*. Other cyclic γ-hydroxyalkenes containing seven- or five-membered rings were well compatible to furnish the corresponding products **4d** and **4e** in 54–79% yields with 88–93% *ee*s. The acyclic γ-hydroxyalkene has a good performance in this reaction, offering the alkoxyalkenylation products **4f-4l** in 43–87% yields with 83–91% *ee*s.

The cyclization of racemic γ-hydroxyalkene as the substrate furnished a 1.1:1 diastereomeric mixture of the product **4 m** in 56% yield with 84% and 93% *ee*. The desymmetrization of mesomeric γ-hydroxyalkene was realized by using **Xu6**, delivering **4n** in 52% yield as a 2:1 mixture of diastereomer with considerable *ee*. The secondary γ-hydroxyalkene also produced a mixture of diastereomers of **4o** in 21% yield with 94% *ee* and 80% *ee*. The relatively lower yield may be attributed to the competitive side oxidation reaction of the alcohol[59].

Further studies showed that a series of products bearing a tertiary stereocenter could be furnished in moderate to good yields with high enantioselectivities from the corresponding methyl-substituted tertiary γ-hydroxyalkenes (Fig. 4). Alkenes with a five- or six-membered ring were well tolerated to give the desired products **6a-6b** with 63–91% yields and 88–90% *ee*s. The reactions with a range of alkenyl bromides bearing a monosubstituted phenyl ring with an electron-donating group (OMe) or an electron-withdrawing group (F) at the *ortho*- or *para*-position furnished the desired products **6c-6e** in 89–93% yields with 90–91% *ee*s. The use of 1-naphthyl alkenyl bromides and 2-naphthyl alkenyl bromides delivered the enantiopure products

**6f-6g** bearing a quaternary stereocenter in 95% yields with 87–91% *ee*s. In addition, the 1,3-dienyl bromide was well compatible under the reaction condition, furnishing **6 h** in 96% yield with 92% *ee*.

It is well known that *Z*-olefins are the basic structural unit of organic molecules and thermodynamic unstable, rendering a jumbo challenge for the synthesis of functionalized *Z*-olefins with highly selectivity[64]. When the *Z*-alkenyl bromide **2b** was utilized in the reaction, the target product *Z*-**3b** could be obtained in 57% yield with 88% *ee* by using **Xu2** as the ligand (Fig. 4). This result indicated the alkene configuration was retained in this process.

After finished enantioselective alkoxyalkenylation reaction of tertiary γ-hydroxyalkenes, we turned to investigate the more challenging primary γ-hydroxy alkenes due to the competitive side reaction, i.e. the oxidation[59] (Fig. 5). Moreover, the enantiocontrol was believed to be more difficult than tertiary alcohols. Indeed, we didn't get good result (63% yield, 67% *ee*) at the beginning under standard conditions. Fortunately, another ligand **Xu9** was identified as a good ligand, delivering a very exciting result (68% yield, 92% *ee*) under the conditions with the 2.5 mol% of [Pd(allyl)Cl]₂ and 10 mol% ligand (**Xu9**), NaOEt as the base, in toluene at 0 °C. The scope of alkenyl bromides with primary γ-hydroxyalkenes was then investigated. A range of alkenyl

bromides were compatible in the reaction. The alkenyl bromides bearing electron-donating groups on the R², such as alkyl group (**8c, 8d, 8e, 8 g**), alkyloxy (**8a, 8h-8k**), phenyl (**8 f**), and amino (**8 m**) coupled with **7** to afford products 44%-73% yields with high enantioselectivities (91–94% *ee*). However, the R² changing to 3,4,5-OMe-Ph (**8 l**), delivered a relatively lower yield, which might be due to the slow oxidation, but the enantioselectivity remains high (94% *ee*). Electron-withdrawing groups (**8n-8p**), thiophenyl (**8q**), and naphthyl (**8r, 8 s**) on the R² also could give satisfactory aim products. In these transformations, α-bromostyrene (**8 u**) and multi-substituted vinyl bromides (**8 v**) could also give good results.

Then, we turned our attention to investigating other primary γ-hydroxyalkenes bearing various substituents at different positions. The reaction with primary γ-hydroxyalkenes with substituted Bn groups at α-position proceeded smoothly to afford products **10a-10h** in moderate to good yields and high *ee*s. Besides Bn group, phenyl (**10j**), allyl (**10i**), and alkyl group (**10k-10n**) were also well tolerated, furnishing the products in 62–78% yields with 87–94% *ee*s. Next, we turned to examine the primary γ-hydroxyalkenes with two substituents at the β-position. The reaction temperature needed to be increased to 0 °C and the desired products **10o** and **10p** were obtained in 72% and

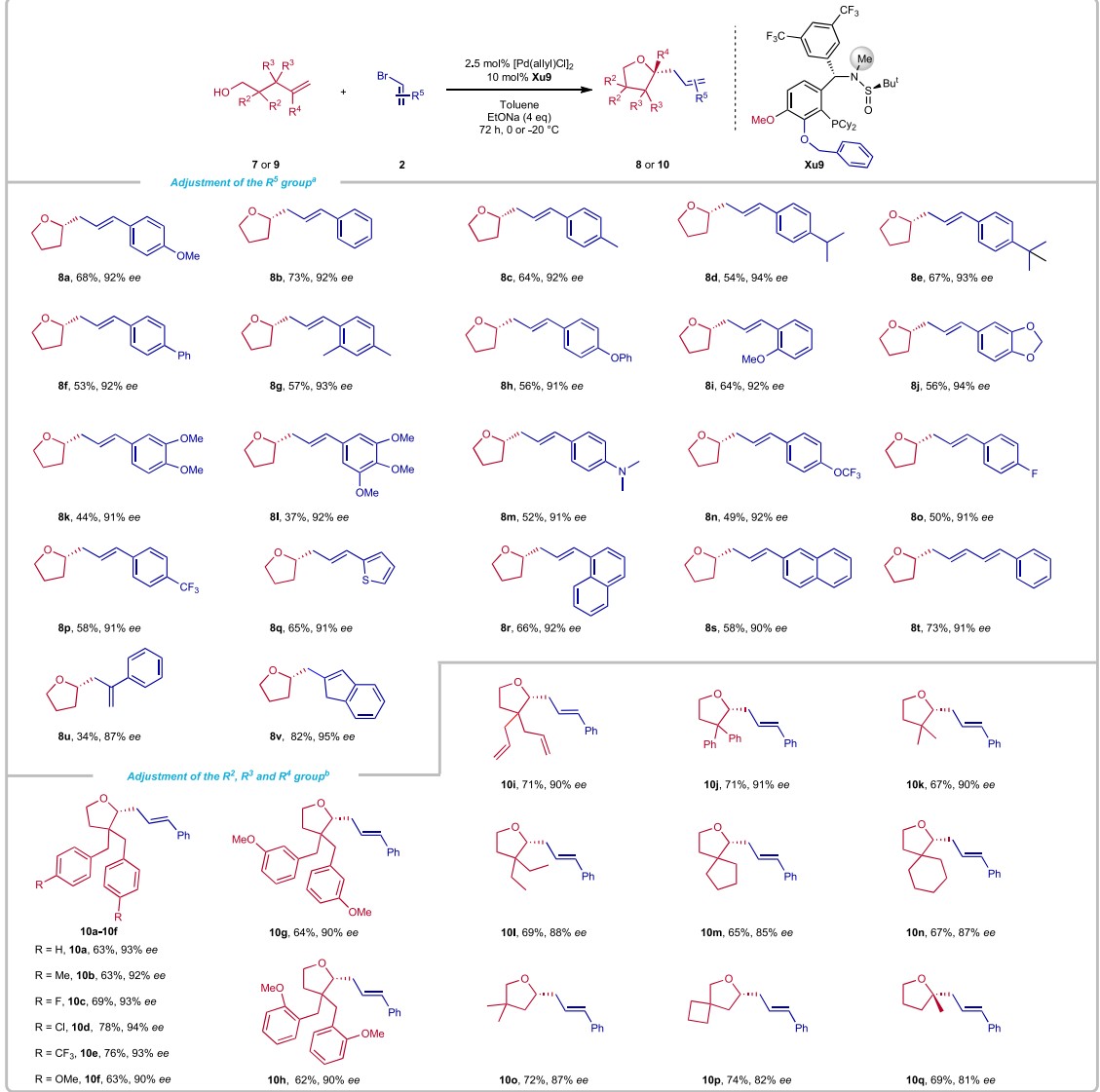

**Fig. 5 | Synthesis of 8 or 10.** Reaction conditions: **7** or **9** (0.5 mmol), **2** (1 mmol), EtONa (4 equiv.), [Pd(allyl)Cl]₂ (2.5 mol%), **Xu9** (10 mol%) in 3.5 mL toluene under N₂ for 72 h. Isolated yields. *ee* was determined by chiral HPLC analysis. [a]at 0 °C. [b]at −20 °C.

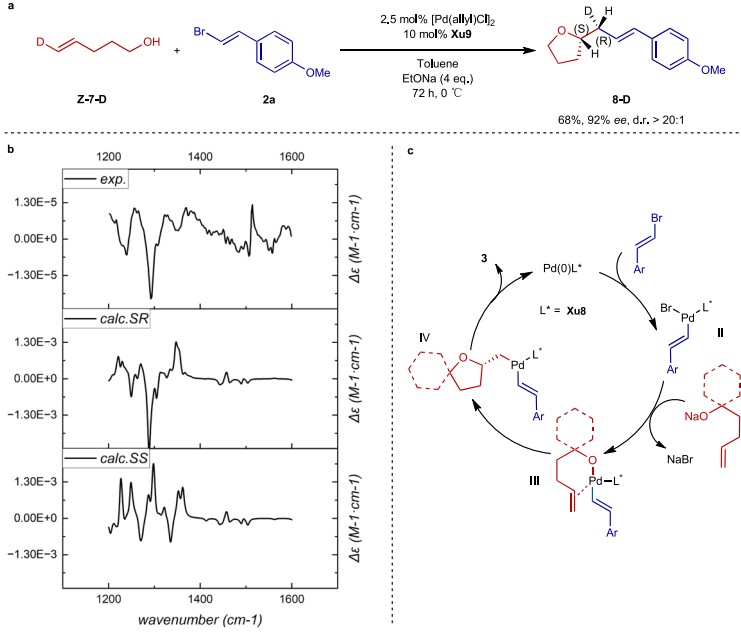

**Fig. 6 | Investigation of the reaction mechanism. a** Deuterium labeling experiment. **b** Comparison of experimental and computational VCD spectra. **c** Proposed catalytic cycle.

74% yield with 87% and 82% *ee*, respectively. Compound **10q** containing a tertiary stereocenter could also be obtained in 69% yield with 81% *ee* from the corresponding primary γ-hydroxyalkene with $R^4$ (methyl) group.

## Mechanistic Investigation

To gain deep insight into the reaction mechanism and the origin of the stereoselectivity, further experimental and computational investigations were carried out. The *cis*- or *trans*-nucleopalladation attack of terminal alkenes has always been ambiguous since both patterns provide the same products. To determine the product's relative stereochemistry, we used the reaction of compound **7** labeled with deuterium (**Z-7-D**) and **2a** with **Xu9** as the ligand (Fig. 6a). The reaction achieved a high d.r. value (>20:1), indicating a highly specific mechanism of the oxypalladation step. Since the absolute configuration of the stereocenter in the THF ring was confirmed as *S* by XRD result (vide infra). We narrowed the possible stereochemistry of product **8** down to *SS* or *SR*. By comparing the experimental and computational (details provided in SI) VCD spectra (Fig. 6b), we found that **8-D** takes an *SR* configuration, which confirms a *cis*-oxypalladation operative mechanism. To the best of our knowledge, this is the first VCD application in the mechanism determination of organometallic reactions.

Based on the verified *cis*-oxypalladation mechanism, the proposed catalytic cycle is shown in Fig. 6c. First, the $Pd^0$ species is transformed into the $Pd^{II}$ complex by the oxidative addition of alkenyl bromide (**2**). In the presence of NaO$^t$Bu and alcohol (**1**), the Pd−O bond is formed to afford complex **III** via ligand exchange. Then **III** proceeds a migratory insertion, delivering the Pd complex **IV**. This is the stereoselectivity-determining step. Through reductive elimination of complex **IV, 3** is obtained, and the catalyst can be regenerated. To further interrogate how our ligands may impart stereoselectivity, DFT calculations were performed. **Xu1, Xu2, Xu3, Xu6, Xu8, Xu9**, and corresponding substrates were taken as subjects. After thorough conformation searches of the migratory insertion TSs (transition states), the calculated *ee* values match well with the experimental results (Supplementary Table 2). The TSs structures give us a way to inspect not only the origin of the high stereoselectivity, but also the unique role of the side-arm in enhancing the stereoselectivity.

Taking **Xu8** for example, three main factors may benefit the stereoselectivity. First, the conformation of the carbon chain of **1a** differs obviously between major and minor TSs. Intuitively, in the minor TS, **1a** interacts with the biphenyl substitution group of **Xu8** more intensively, causing the scaffold of the ligand to move correspondingly, resulting in some instability. Second, the conformational changes further affect the side-arm (OBn), and a significant conformational change of the side-arm is witnessed (Fig. 7a). In the major TS, a CH-π interaction is formed between the side-arm benzyloxy group and the vinyl substrate (shown as a green isosurface in Fig. 7a), which is absent in the minor TS. This stabilizes the major TS and benefits stereoselectivity (92% *ee*). Similar CH-π interaction and difference between major and minor TSs can be found in **Xu9** as well.

To disentangle possible controlling effects and provide quantitative proof for the analysis above, the energy difference ΔΔE between the major and minor TSs of **Xu6** and **Xu8** was decomposed. We compared the results of **Xu6** and **Xu8** and found that the energy difference between TSs mainly came from the dimer of Xu-Phos and vinyl substrate (Supplementary Table 4). Further decomposition of the energy indicated that the interaction between Xu-Phos and vinyl substrate is the main contributor (Fig. 7b). We investigated the strength of CH-π interactions between the side-arm benzyloxy group and vinyl substrate. Our computational analysis revealed an interaction energy of 3.6 kcal/mol (Fig. 7c)[65], which closely aligns with the Xu-Phos-vinyl substrate interaction energy difference between **Xu6** and **Xu8** (3.4 kcal/mol).

Furthermore, we observed that the substituent group *ortho* to the side-arm plays a crucial role in the side-arm effect. Specifically, in **Xu6**, the OBn group remained in the same position (resting position) in both TSs, and consequently had no impact on stereoselectivity (82% *ee*). However, when OMe was introduced in **Xu8**, the restricted space forced OBn to turn away from the resting position, leading to distinct conformations and, as a result, a more significant energy difference between the major and minor TSs. We conducted further theoretical and experimental analysis by substituting OMe with OEt, O$^i$Pr, and O$^t$Bu groups (Supplementary Table 6), which revealed that bulky groups (i.e. O$^i$Pr and O$^t$Bu) will interact with vinyl substrate instead of side-arm in minor TSs and reduced the energy difference between major and minor TSs. Consequently, it becomes evident that an

**a**

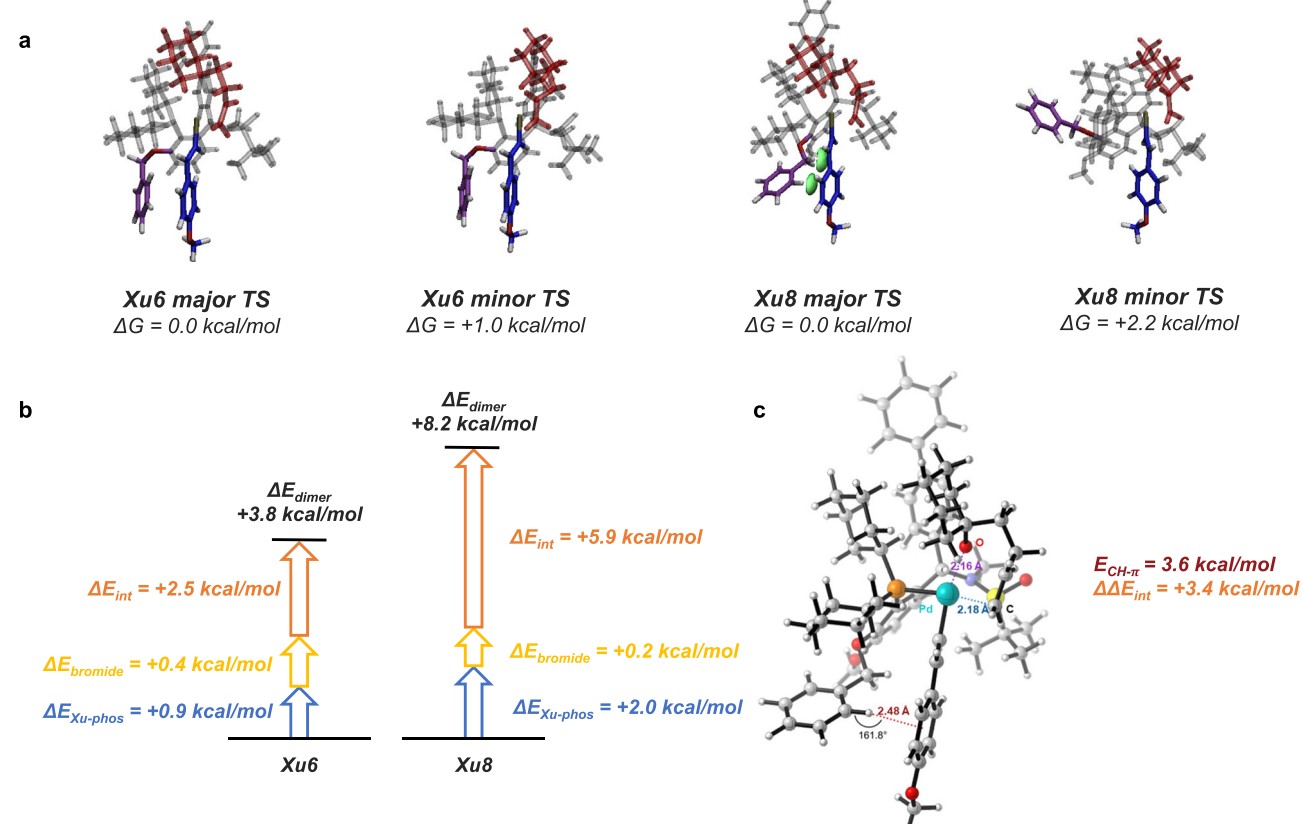

*Xu6 major TS*
*ΔG = 0.0 kcal/mol*

*Xu6 minor TS*
*ΔG = +1.0 kcal/mol*

*Xu8 major TS*
*ΔG = 0.0 kcal/mol*

*Xu8 minor TS*
*ΔG = +2.2 kcal/mol*

**b**

$ΔE_{dimer}$ +8.2 kcal/mol

$ΔE_{dimer}$ +3.8 kcal/mol

$ΔE_{int}$ = +2.5 kcal/mol

$ΔE_{int}$ = +5.9 kcal/mol

$ΔE_{bromide}$ = +0.4 kcal/mol

$ΔE_{bromide}$ = +0.2 kcal/mol

$ΔE_{Xu-phos}$ = +0.9 kcal/mol

$ΔE_{Xu-phos}$ = +2.0 kcal/mol

Xu6          Xu8

**c**

$E_{CH-π}$ = 3.6 kcal/mol
$ΔΔE_{int}$ = +3.4 kcal/mol

**Fig. 7 | Visualization study and energy decomposition of the TSs. a** Optimized TS structures for Pd/**Xu8** and Pd/Xu6 catalyzed reactions of 1a and 2a. CH-π interaction between side-arms and substrates in **Xu8** major is shown as green isosurfaces. **b** Energy difference decomposition result of Xu-Phos·vinyl substrate dimers. ΔE is calculated as minor structure energy subtracted major structure energy. $E_{Xu-Phos}$ represents the energy of Xu-Phos, $E_{vinyl}$ represents the energy of the vinyl substrate,

and $E_{int}$ represents the interaction energy between Xu-Phos and the vinyl substrate. **c** The core structure and CH-**π** interaction of **Xu8** major TS. CH-**π** interaction is denoted with red dashed lines. The value of approximate intensity of the CH-**π** interaction and the difference of interaction energy between dimers in **Xu6** and **Xu8** are also shown.

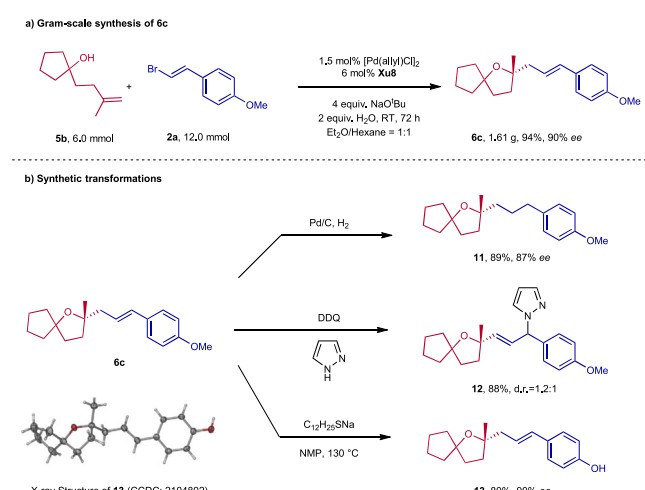

**a) Gram-scale synthesis of 6c**

5b, 6.0 mmol     +     2a, 12.0 mmol

1.5 mol% [Pd(allyl)Cl]₂
6 mol% **Xu8**
4 equiv. NaOᵗBu
2 equiv. H₂O, RT, 72 h
Et₂O/Hexane = 1:1

6c, 1.61 g, 94%, 90% *ee*

**b) Synthetic transformations**

Pd/C, H₂ → 11, 89%, 87% *ee*

6c

DDQ / pyrazole → 12, 88%, d.r.=1.2:1

X-ray Structure of 13 (CCDC: 2104802)

C₁₂H₂₅SNa / NMP, 130 °C → 13, 80%, 90% *ee*

**Fig. 8 | Gram-scale synthesis and transformations of products. a** Gram-scale synthesis of 6c. **b** Synthetic transformations of 6c.

appropriate group situated adjacent to the side-arm must serve a dual purpose. Not only should it impart steric hindrance, but it should also strike a balance by avoiding excessive bulkiness that might trigger undesired interactions with the vinyl substrate.

To showcase the synthetic application of this methodology, a gram-scale reaction and several transformations of **6c** were carried out

(Fig. 8). Under the catalysis of 1.5 mol% [Pd(allyl)Cl]₂ at RT, 1.61 g of **6c** was obtained in 94% yield with 90% *ee*. A hydrogenation reaction of **6c** delivered the corresponding product **11**. Treatment of **6c** with 1H-pyrazole under the oxidation of DDQ led to compound **12** with 1.2:1 dr. The enantiopure **6c** could undergo a demethylation reaction to produce compound **13**, whoseDiscussion.

In summary, we have developed an efficient palladium-catalyzed enantioselective alkoxyalkenylation of a range of primary, secondary, and tertiary γ-hydroxyalkenes with alkenyl halides. This reaction provides facile access to a series of enantiopure tetrahydrofurans containing an olefin framework with a tertiary or quaternary stereocenter. Notably, this process features a broad substrate scope, good functional group tolerance, and mild reaction conditions. Through a combination of experimental and DFT calculation investigation, we verified the *cis*-oxypalladium mechanism and located the stereo-determining TSs. Additionally, the introduction of a side-arm to the chiral ligand is a crucial strategy for achieving high efficiency and enantioselectivity. These findings offer exciting opportunities in asymmetric reactions, and the investigation of the side-arm effect may help in the rational design of chiral ligands.

## Methods
### General procedure for synthesis of 3, 4, or 6
To a sealed tube was added [Pd(allyl)Cl]₂ (2.5 mol%), **Xu8** (10 mol%). The flask was evacuated and refilled with argon. Then tertiary γ-hydroxyalkenes (0.2 mmol), alkenyl halides (0.4 mmol), NaOᵗBu (4.0 equiv.), H₂O (2.0 equiv.) and a mixed solution of Et₂O/Hexane (1: 1,

2 mL) was added to the tube and stirred at room temperature for 72 h. After the reaction was complete (monitored by TLC), the solvent was removed under reduced pressure. The crude product was then purified by flash column chromatography on silica gel to afford the desired product **3, 4**, or **6**.

## General procedure for synthesis of 8 or 10

To a sealed tube was added [Pd(allyl)Cl]₂ (2.5 mol%), **Xu9** (10 mol%). The flask was evacuated and refilled with argon. Then γ-hydroxyalkenes (0.5 mmol), alkenyl halides (1 mmol), EtONa (4.0 equiv), and solution of toluene (3.5 mL) was added to the tube and stirred at 0 °C or -20 °C for 72 h. After the reaction was complete (monitored by TLC), the solvent was removed under reduced pressure. The crude product was then purified by flash column chromatography on silica gel to afford the desired product **8** or **10**.

## Data availability

The data that support the findings of this study are available within the paper and its Supplementary Information files. The X-ray crystallographic coordinates for structures reported in this study have been deposited at the Cambridge Crystallographic Data Centre (CCDC), under deposition numbers 1977366 (**Xu6**-borane complex) and 2104802 (13). These data can be obtained free of charge from The Cambridge Crystallographic Data Centre via www.ccdc.cam.ac.uk/data_request/cif. The coordinates of DFT calculated structures are provided in Supplementary Information file. All data are also available from authors upon request.

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

## Acknowledgements

We gratefully acknowledge the funding support of NSFC (22031004 and 21921003 for J.Z., and 22071060 for W.L.), National Key R&D Program of China (2021YFF0701600 for J.Z.) Shanghai Municipal Education Commission (20212308 for J.Z.), and Shanghai Science and Technology Commission (23ZR1404800 for Z.L.).

## Author contributions

J.Z. conceived the project. J.Z., Z.L., S.Z., M.C., Z.Y. and W.L. wrote the paper. S.Z. and M.C. performed the experiments and analyzed the data. Z.Y. and Z.L. conducted the computational studies. L.W., Z.W., K.Z., and H.D. analyzed the data. All authors discussed the results and commented on the paper.

## Competing interests

The authors declare no competing interests.
