## [Peer Review File · Nature Communications]

REVIEWER COMMENTS

Reviewer #1 (Remarks to the Author):

Although the intramolecular alkoxy-alkenylation of alkene was already reported by the same group in <https://doi.org/10.1002/anie.201912408>, with very similar Xiang-Phos ligand, I still recognize the improvement of this method in using alcohols as reactants, especially that this method can be applied to primary, secondary, and tertiary alcohols. Besides, VCD application in the mechanism determination of organometallic reactions is very interesting. I suggest it could be accepted after addressing the following comments.

Entry 14 in table 1 exhibited a nearly similar best reaction condition, with even a better yield. I strongly suggested the authors to utilize this condition to improve the yields of those reactions exhibiting moderate or poor yields in Figures 4 and 5.

The push effect of the side-arm ligand is still not so solid, even though with DFT calculations. It is better to modify the MeO group to EtO group, *i*PrO or BuO group, and further verify the concept by the combination of experimental results, as well as the obtained theoretical results.

With respect to the VCD application in the mechanism determination of organometallic reactions is very interesting, the authors should also provide the calculated RS and RR VCD spectrum for comparison.

How did the authors calculate the energy value of the V the CH- π interaction? This should be clearly described in the computational method section.

It is strange that the conformation of OBn has such a big change between Xu8 major TS and Xu8 minor TS. Note that the conformations in Xu6 major TS and Xu6 minor TS should not lead to such a big difference when replacing the adjacent H to OMe in in Xu6 major TS and Xu6 minor TS. It should be very carefully to avoid any errors due to the wrong conformation. The authors should 1) recalculate the Xu8 major TS and Xu8 minor TS by replacing the adjacent H to OMe based on the structures of Xu6 major TS and Xu6 minor TS; 1) recalculate Xu6 major TS and Xu6 minor TS by altering the OMe group to H atom on the basis of the structures of Xu8 major TS and Xu8 minor TS,

and make a carefully analysis of these results of the eight transition states (by analyzing both the conformations and the energy difference)

Reviewer #2 (Remarks to the Author):

The manuscript under consideration details the development of a Pd-catalyzed enantioselective alkoxyalkenylation of γ -hydroxyalkenes facilitated by ligand design. This catalytic protocol is efficient and general, as demonstrated by the broad and extensive substrate scope. In-depth mechanistic studies have been conducted to clarify the reaction mechanism and the role of the newly designed chiral ligand (side-arm effect) on the enantio- and diastereoselectivity of the reaction. The current asymmetric Alkoxyalkenylation is considered to be a significant extension of Pd-catalyzed carboetherification reaction, especially using alkenyl bromides as substrates. Therefore, this study is recommended as a potential publication in Nature Communications after addressing the following comments.

1. While the structure of ligands are influential to the success of the current reaction, other reaction parameters, including solvent, Pd precursors and additives, are also demonstrated to be important by the authors in Table 1. In particular, the addition of water was found to be crucial to obtaining high enantioselectivity. Can the authors explain or clarify the role of H₂O in the catalytic process?

2. While the calculated TSs demonstrate that only one ligand ligated to Pd during the catalytic process, a Pd:L ratio of 1:2 was adopted for all reactions. Will comparable reactivity and stereoselectivity be achieved if an 1:1 Pd:L ratio is employed?

3. the utilization of the English language with regards to grammar and form is currently not at the state for publication and the authors are encouraged to double-check and go through the whole manuscript. Several examples are shown below:

-Introduction section, the secondary paragraph: The first sentence is not complete.

- Introduction section, the third paragraph: 1) "...aryl bromine in the reaction is very different," should be corrected as "...aryl bromide in the reaction are very different,"; 2) "which remains as a unsolved problem and pose considerable challenges" should be revised as "which remains as an unsolved problem and poses considerable challenges"; 3) the last sentence, "were located and reveal".

-Reaction Development section, the last two sentences of paragraph two: "yieled" should be corrected as "yielded"

-Discussion section, "provids" should be revised as "provides"

4. Signals (peaks) from impurities are observed in the ¹H NMR spectra of compound 10e and 10i. The authors are encouraged to further purify these samples if possible.

Reviewer #3 (Remarks to the Author):

This manuscript details a novel palladium-catalyzed enantioselective alkoxyalkenylation reaction of a range of primary, secondary, and tertiary γ -hydroxyalkenes with alkenyl halides and details two newly identified sulfinamide phosphine (Xu-Phos) ligands. The authors use VCD and DFT to verify the cis-oxypalladation mechanism and located the stereo-determining TSs. In addition, the newly identified Xu-Phos ligands are structurally characterized using single-crystal X-ray diffraction techniques.

While my expertise lies in crystallography, I believe this work does make a significant contribution to the field and should be published given minor revisions as outlines below:

There are several issues with the crystal structures that the authors should fix and/or comment on:

exp_801: In general, the authors need to respond to all B-, and C-level alerts present in the CIF report. There is no temperature record (293 K) and the authors should state the temperature at which data was collected. If this alert is correct, the authors should comment on why data was not collected at low temperature. There are several alerts relating to missing FCF reflections and numerous omitted reflections (31) in the data set. The authors need to state why these reflections are being omitted and if there is not a good reason, these reflections should be included in the structure analysis.

exp_2079: In general, the authors need to respond to all C-level alerts present in the CIF report. There are several alerts relating to missing FCF reflections and numerous omitted reflections (17) in the data set. The authors need to state why these reflections are being omitted and if there is not a good reason, these reflections should be included in the structure analysis.

In both crystal structures, there is chirality noted and the authors need to verify if the given chirality is correct and note how this was determined (e.g. known chirality from starting material, anomalous dispersion effects, etc.). Given some of the alerts, it would be good for the authors to look for missing solvent or twinning effects and, if none are found, state this in the manuscript along with responding appropriately to the alerts in the CIF report.

TITLE: Palladium/Xu-Phos-Catalyzed Enantioselective Alkoxyalkenylation of γ -Hydroxyalkenes: Side Arm Effect and Mechanistic Study

Manuscript ID: NCOMMS-23-17932

Authors: Shuai Zhu, Zihao Ye, Ming-Jie Chen, Lei Wang, Yu-Zhuo Wang, Ke-Nan Zhang, Wen-Bo Li, Han-Ming Ding, Zhiming, and Junliang Zhang

Dear editor

As you have requested, I reviewed the VCD analysis performed in this study.

To gain insight into the reaction mechanism and the origin of stereo-selectivity, the authors have considered as example the reaction described in Figure 6 of the main manuscript. By combining deuterium labelling with vibrational circular dichroism (VCD) spectroscopy, they have assigned the configuration of the resulting compound (8-D) to be SR. This in turn, confirmed a cis-oxypalladation operative mechanism.

The assignment of the configuration of the 8-D compound is based on the observation that in the frequency interval between 1200 and 1400 1/cm, the VCD spectra computed for the SR and SS configurations of 8-D have bands of opposite signs. Since the spectrum computed for the SR configuration resembles the experiment better than the SS spectrum, the authors have concluded that the configuration of 8-D is SR. The procedure is correct and standard.

Since the 8-D molecule is small I have performed test calculations for several 8-D-SR and 8-D-SS conformers. I can confirm that many of the conformers have in the SR and SS configurations VCD bands of opposite sign in the frequency interval between 1220 and 1330 1/cm. **Therefore, I believe that the assignment made by the authors is correct.** To be hundred percent sure, I ask the authors to provide some additional information and perform a few additional tests:

- 1) Show in the ESI also the comparison between the experimental and simulated IR spectra. To properly judge the relative sign of the SR and SS VCD bands and how similar they are with the experimental spectrum, the associated IR spectra need to be aligned with the experimental one.
- 2) Investigate how sensitive the prediction made with VCD is to a change of computational parameters. I am asking this because the simulated IR/VCD spectra are computed as a Boltzmann average of the spectra computed for the individual conformers, and the Boltzmann factors computed with DFT are inaccurate and very sensitive to the choice of computational parameters.

Therefore, since not all conformers seem to exhibit VCD bands of opposite sign in the interval between 1200 and 1400 1/cm, it is important to verify how the SR and SS Boltzmann weighted VCD spectra change when using Boltzmann factors predicted by different levels of theory. Since the experiment was performed in CDCl₃, I would suggest standard vacuum calculations (i.e., without dispersion correction and without specifying the chloroform as solvent) using the 6-31G (d, p) and def2-TZVP basis sets. This additional analysis is not expensive and should increase the level of confidence for the prediction made here with VCD.

3) Show in the ESI comparisons of the overlaid SR and SS VCD spectra of the considered low-energy conformers.

Some additional points:

1) There seems to be an inconsistency regarding the compounds used in the VCD analysis. Section 6 in ESI mentions that the IR and VCD measurements were performed for compound 8-D, which is in line to what is written in the main manuscript. On the other hand, in Section 7 in ESI it is stated that the IR and VCD calculations were performed for the 3a-D-SS and 3a-D-SR compounds. I assume this is a typo, however, the authors need to clarify/correct this point. Obviously, the calculations should be performed for the molecule on which the measurement was performed.

2) When describing the computational VCD procedure, the authors mention that conformational searches have been performed for both SR and SS configurations. This of course is not necessary, the SR and SS structure are obtained by simply exchanging a hydrogen atom with a deuterium atom, so one expects identical results from the two conformational searches. More worrying is, however, the fact that only 11 low-energy conformers have been found within an energy window of 2 kcal/mol. The very basic conformational search I performed with RDKit has yielded 32 low-energy conformers for 8-D (which have different VCD spectra) within that same energetic window. Therefore, I am wondering whether the authors have considered all relevant conformers. What program has been used when performing the conformational search? The citations given by the authors (i.e, 22-24) don't seem to be correct as they do not describe a conformation search procedure/program.

3) There are many errors associated with the citations in Section 7 in the ESI (i.e. on page S115). For example: ¹⁰(75), ¹¹(76),¹²⁻¹³(77, 78), etc.

Sincerely your,
Dr. Paul Nicu

Response to Reviewer 1:

Comment: Although the intramolecular alkoxy-alkenylation of alkene was already reported by the same group in <https://doi.org/10.1002/anie.201912408>, with very similar Xiang-Phos ligand, I still recognize the improvement of this method in using alcohols as reactants, especially that this method can be applied to primary, secondary, and tertiary alcohols. Besides, VCD application in the mechanism determination of organometallic reactions is very interesting. I suggest it could be accepted after addressing the following comments.

1. Entry 14 in table 1 exhibited a nearly similar best reaction condition, with even a better yield. I strongly suggested the authors to utilize this condition to improve the yields of those reactions exhibiting moderate or poor yields in Figures 4 and 5.

Reply: Thank you for your constructive advice. We used toluene as a solvent (entry 14 in table 1) to investigate 3c, 3g, 3i, 3j, 3l, 4d, 4g, 4i, 4l in Figure 4. Analysis of the outcomes revealed enhanced yields for 3c, 3i, and 3j, albeit with a decrease in enantioselectivity. Meanwhile, the remaining substrates did not exhibit improvements over the standard conditions. Contrary to the scenarios depicted in Figure 4, the

ligand utilized in Figure 5 was **Xu9**. Notably, optimal conditions were achieved using toluene as the solvent. However, when NaOtBu was employed as the base, worse results were observed (yield: 24%, ee: 87%). As a result, we refrained from

conducting further investigations into the substrates outlined in Figure 5.

2. The push effect of the side-arm ligand is still not so solid, even with DFT calculations. It is better to modify the MeO group to EtO group, iPrO, or BuO group, and further verify the concept by the combination of experimental results, as well as the obtained theoretical results.

*Reply: Thank you for your advice. We have replaced the OMe group in **Xu8** with OEt, OiPr groups respectively to generate new ligands **Xu12** and **Xu13**. The synthesis of **Xu12** and **Xu13** and corresponding Palladium-Catalyzed Enantioselective Alkoxyalkenylation experiments were shown briefly in the following scheme. Detailed synthetic routes can be found in SI. However, it's important to note that the experimental data for **Xu14** with OtBu is currently unavailable due to challenges encountered during its synthesis process.*

Theoretical results and analysis are provided in the 'Analysis of the effect of OR groups ortho to the side-arm' section in SI:

*We originally proposed that the main reason of OMe promoting the side-arm effect is its steric influence. To verify this proposal, the OMe group was replaced with OEt, OiPr, and OtBu groups respectively to generate new ligands **Xu12**, **Xu13**, and **Xu14**. The differences of free energies and ee% values were calculated according to the standard procedure and provided in **Table S6**.*

Table S6. Calculated and experimental ee% of ligands with different OR groups

Entry	Ligand and reactants	Calc. $\Delta\Delta G$ (kcal/mol)	Calc. ee%	Exp. ee%
1	Xu12 + 1a + 2a	2.16	94	90
2	Xu13 + 1a + 2a	0.99	68	85
3	Xu14 + 1a + 2a	1.08	72	/*

* Multiple attempts to synthesize **Xu14** all failed, so experimental ee% value of **Xu14** is not available.

*The high calc. and exp. ee% value of **Xu12** supports that the steric effect of OR groups is beneficial to the improvement of ee%. However, we also found that when the*

OR group gets bulkier, the ee% value starts to drop. From visual inspection of the TSs of Xu13 and Xu14, we suspect that the OR groups might have weak interaction with the vinyl substrate (Figure S7). To verify this, interaction energies between OR groups and vinyl substrates are calculated and provided in Table S7.

Figure S7. Structure of Xu13 and Xu14 TSs

**Table S7. Weak interaction energy between different OR groups and vinyl substrate
(kcal/mol)**

Entry	Ligand and reactants	Eint	ΔE_{int}	Calc. $\Delta\Delta G$
1	Xu8-major	-0.01	/	/

2	Xu8-minor	-0.13	-0.12	2.23
3	Xu12-major	-0.03	/	/
4	Xu12-minor	-0.17	-0.14	2.16
5	Xu13-major	-0.07	/	/
6	Xu13-minor	-0.77	-0.70	0.99
7	Xu14-major	-0.56	/	/
8	Xu14-minor	-1.48	-0.91	1.08

From **Table S7**, it is clear that when the OR group is OMe or OEt, the interaction between the OR group and the vinyl substrate is quite weak and negligible, but when it comes to OiPr and OtBu, the interaction becomes significant. More importantly, the interactions in the minor TSs are stronger than those in the major TSs due to the conformation difference. And ΔE_{int} accounts for the major part of the decrease of the $\Delta\Delta G$ and thus the decrease of ee%.

In conclusion, we posit that the enhancement in ee% achieved by introducing the OMe group into the ortho position of the side-arm is primarily attributed to its steric effect. Nonetheless, the introduction of bulkier substitution groups is likely to lead to weak interactions with the vinyl substrate, particularly in the minor transition state, consequently resulting in a decrease in ee%.

3. Concerning the VCD application in the mechanism determination of organometallic reactions is very interesting, the authors should also provide the calculated RS and RR VCD spectra for comparison.

Reply: Thank you for your advice, we have now provided the VCD spectra of 8-D-RR and 8-D-RS in Figure S5.

We have observed significant disparities in the figures, notably the presence of a distinctive negative peak at approximately 1375 cm^{-1} exclusively in the 8-D-R VCD spectra. Additionally, there is a noticeable difference in relative peak intensities between the two negative peaks around 1300 cm^{-1} in 8-D-SR and 8-D-RR spectra.

Furthermore, it is worth mentioning that we have conducted XRD experiments to validate the absolute configuration of the stereocenter within the THF ring, confirming its *S* configuration (as illustrated in Figure 8, X-ray structure of 13).

Figure S5. Comparison between 8-D-S (left) and 8-D-R (right) VCD spectra.

4. How did the authors calculate the energy value of the V the CH- π interaction?

This should be clearly described in the computational method section.

Reply: Thank you for the suggestion. We have now provided the calculation equation and standard procedure of the interaction energy in SI:

Quantitative calculation of weak interaction

The interaction energy values are defined as the single point energy gap between the complex and components of the complex (Eq 1).²⁹

$$E_{\text{int}} = E_{\text{complex}} - E_{\text{component1}} - E_{\text{component2}} \quad (\text{Eq 1})$$

To illustrate the calculation procedure, we will use the example of calculating the interaction energy between the side-arm and the vinyl substrate in Xu8-ts. Firstly, we isolate the side-arm (OBn group, atom index 77-91) and the vinyl substrate (atom index 108-126) from the TS structure. Next, we introduce a hydrogen atom at each break point of the covalent bonds within these isolated molecules. The positions of

*these added hydrogen atoms are then optimized using standard optimization parameters (as outlined in the General Calculation Procedure), with all other atoms held strictly constrained. Finally, we perform a single-point energy calculation using standard parameters (as specified in the **General Calculation Procedure**). This calculation yields the energy of the complex as well as the individual energies of its constituent parts.*

*Ref 29. Mackenzie W. K., Christopher R. T., Ga Young Lee, Hannah J. E., K. N. Houk, and Marcey L. Waters. More Than π - π - π Stacking: Contribution of Amide- π and CH- π Interactions to Crotonyllysine Binding by the AF9 YEATS Domain. *J. Am. Chem. Soc.* **142**, 40, 17048–17056 (2020).*

5. It is strange that the conformation of OBn has such a big change between Xu8 major TS and Xu8 minor TS. Note that the conformations in Xu6 major TS and Xu6 minor TS should not lead to such a big difference when replacing the adjacent H to OMe in Xu6 major TS and Xu6 minor TS. It should be very careful to avoid any errors due to the wrong conformation. The authors should 1) recalculate the Xu8 major TS and Xu8 minor TS by replacing the adjacent H to OMe based on the structures of Xu6 major TS and Xu6 minor TS; 1) recalculate Xu6 major TS and Xu6 minor TS by altering the OMe group to H atom based on the structures of Xu8 major TS and Xu8 minor TS, and make a careful analysis of these results of the eight transition states (by analyzing both the conformations and the energy difference)

Reply: Thank you for your advice. We generated new conformations according to the procedure you proposed and decomposed the energy to gain deeper insight into the energy difference. Detailed results and analysis are provided in the ‘Analysis of different conformation of Xu6 and Xu8 TSs’ section in SI:

*We proposed that the side-arm conformations of **Xu6** and **Xu8** are critical to the enantioselectivity. To certify that current conformations are not a result of chance, we replaced H adjacent to the side-arm in **Xu6** TSs to OMe to generate new conformation of **Xu8** TSs and also generated new conformation of **Xu6** TSs with a similar procedure. The energy difference profile of the TSs is provided in **Table S5**, the*

structures are shown in **Figure S6**, and coordinates are provided in **Table S8**.

Table S5. The energy difference between conformations of Xu6 and Xu8 TSs (kcal/mol)

TS name	$\Delta E_{\text{Xu-phos}}$	ΔE_{other}	$\Delta E_{\text{int-sidearm}}$	$\Delta E_{\text{int-all}}$	ΔE_{all}	ΔG_{all}
Xu6-major	2.51	1.55	-0.04	-3.08	0.98	0.66
Xu6-minor	-1.50	0.06	3.67	3.83	2.39	4.07
Xu8-major	0.36	-0.08	0.02	-0.39	-0.10	1.42
Xu8-minor	1.90	-0.28	-3.03	-0.50	1.12	2.08

- 1) All energy differences are calculated as the new TSs' energy subtract the original TSs' energy
- 2) $E_{\text{Xu-phos}}$ refers to the single point energy of the **Xu-phos** segment in the TS structures
- 3) E_{other} refers to the sum of single point energy of all segments except for **Xu-phos** (i.e. vinyl substrate and alcohol substrate) in the TS structures
- 4) $E_{\text{int-sidearm}}$ refers to the interaction energy between the side-arm and the vinyl substrate in the TS structures
- 5) $E_{\text{int-all}}$ refers to the sum of the interaction energies between all segments in the TS structures
- 6) E_{all} refers to the single-point energy of the TS structures
- 7) G_{all} refers to the free energy of the TS structures

Figure S6. Structures of different structures of Xu6 and Xu8 TSs.

Referring to **Table S5**, we can make the following observations: 1) None of the new TS structures exhibit lower free energy compared to their corresponding original TS structures. This reinforces our analysis, which is primarily based on the original TS structures; 2) Focusing on the **Xu6**-major TSs, a noteworthy increase in energy,

denoted by $\Delta E_{\text{Xu-phos}}$, suggests that the introduction of the OMe group does indeed compel the side-arm to adopt a different position (even though both positions are downward), resulting in an overall higher energy for the entire ligand (+2.51 kcal/mol); 3) Concerning the **Xu6**-minor TSs, it is notable that the primary contributor to the increase in energy (+3.67 kcal/mol) is the loss of interaction between the side-arm and the vinyl substrate. This underscores the significance of this interaction in the overall energetics of the system; 4) In the case of **Xu8**-major TSs, after the H was substituted with OMe, the side-arm was optimized spontaneously to a conformation similar to the original **Xu8**-major TS. As a result, the single-point energies of these structures turned out to be very similar.; 5) Concerning **Xu8**-minor TSs, forcing the side-arm to take the downward conformation indeed makes the ligand energy higher (+1.90 kcal/mol), which is the main contributor to the overall energy rise, and although the interaction between the side-arm and vinyl substrate is stronger, it is masked by the loss of other interactions.

Response to Reviewer 2:

The manuscript under consideration details the development of a Pd-catalyzed enantioselective alkoxyalkenylation of γ -hydroxyalkenes facilitated by ligand design. This catalytic protocol is efficient and general, as demonstrated by the broad and extensive substrate scope. In-depth mechanistic studies have been conducted to clarify the reaction mechanism and the role of the newly designed chiral ligand (side-arm effect) on the enantio- and diastereoselectivity of the reaction. The current asymmetric Alkoxyalkenylation is considered to be a significant extension of Pd-catalyzed carboetherification reaction, especially using alkenyl bromides as substrates. Therefore, this study is recommended as a potential publication in Nature Communications after addressing the following comments.

1. While the structure of ligands are influential to the success of the current reaction, other reaction parameters, including solvent, Pd precursors and additives, are also demonstrated to be important by the authors in Table 1. In particular, the addition

of water was found to be crucial to obtaining high enantioselectivity. Can the authors explain or clarify the role of H₂O in the catalytic process?

*Reply: Thank you for your constructive advice. Because we used 4 equivalents of NaOtBu and 2 equivalents of H₂O, it will generate the NaOH, and tBuOH and a mixture bases were formed. Water or mixture of bases have been reported to have the potential to improve enantioselectivity in Pd-catalyzed reactions. Nonetheless, the precise mechanism underlying how water enhances enantioselectivity in the catalytic process remains unclear. (Angew. Chem. Int. Ed. **2012**, 57, 9886-9890. J. Am. Chem. Soc. **2003**, 125, 14133-14139.).*

2. While the calculated TSs demonstrate that only one ligand ligated to Pd during the catalytic process, a Pd:L ratio of 1:2 was adopted for all reactions. Will comparable reactivity and stereoselectivity be achieved if an 1:1 Pd:L ratio is employed?

Reply: Thank you for your constructive advice. A 1:1 Pd:L ratio is employed, we got 3a with 44% yield and 86% ee.

3. The utilization of the English language with regards to grammar and form is currently not at the state for publication and the authors are encouraged to double-check and go through the whole manuscript. Several examples are shown below:

-Introduction section, the secondary paragraph: The first sentence is not complete.

-Introduction section, the third paragraph: 1) "...aryl bromine in the reaction is very different," should be corrected as "...aryl bromide in the reaction is very different,"; 2) "which remains as a unsolved problem and pose considerable challenges" should be revised as "which remains as an unsolved problem and poses considerable challenges"; 3) the last sentence, "were located and reveal".

-Reaction Development section, the last two sentences of paragraph two: “yieled” should be corrected as “yielded”

-Discussion section, “provids” should be revised as “provides”

Reply: Thank you for your constructive advice. We have double-checked and gone through the whole manuscript, and the manuscript has been updated.

4. Signals (peaks) from impurities are observed in the ¹H NMR spectra of compound 10e and 10i. The authors are encouraged to further purify these samples if possible.

Reply: Thank you for your constructive advice. The two samples have been purified, and the ¹H NMR spectra of them have been updated in SI.

Response to Reviewer 3:

This manuscript details a novel palladium-catalyzed enantioselective alkoxyalkenylation reaction of a range of primary, secondary, and tertiary γ -hydroxyalkenes with alkenyl halides and details two newly identified sulfinamide phosphine (Xu-Phos) ligands. The authors use VCD and DFT to verify the cis-oxypalladation mechanism and located the stereo-determining TSs. In addition, the newly identified Xu-Phos ligands are structurally characterized using single-crystal X-ray diffraction techniques.

While my expertise lies in crystallography, I believe this work does make a significant contribution to the field and should be published given minor revisions as outlines below:

There are several issues with the crystal structures that the authors should fix and/or comment on:

1. exp_801: In general, the authors need to respond to all B-, and C-level alerts present in the CIF report. There is no temperature record (293 K) and the authors should state the temperature at which data was collected. If this alert is correct, the authors should comment on why data was not collected at low temperature. There are several alerts relating to missing FCF reflections and numerous omitted reflections

(31) in the data set. The authors need to state why these reflections are being omitted and if there is not a good reason, these reflections should be included in the structure analysis.

Reply: Thank you for your constructive advice. Data was collected at 100K. Low temperatures are used to obtain better diffraction and to determine the absolute configuration. Only three relatively poor diffraction points at low angles are omitted, probably because of the beam stop.

2. exp_2079: In general, the authors need to respond to all C-level alerts present in the CIF report. There are several alerts relating to missing FCF reflections and numerous omitted reflections (17) in the data set. The authors need to state why these reflections are being omitted and if there is not a good reason, these reflections should be included in the structure analysis.

Reply: Thank you for your constructive advice. Both CIF and CheckCIF files have been updated.

3. In both crystal structures, there is chirality noted and the authors need to verify if the given chirality is correct and note how this was determined (e.g. known chirality from starting material, anomalous dispersion effects, etc.). Given some of the alerts, it would be good for the authors to look for missing solvent or twinning effects and, if none are found, state this in the manuscript along with responding appropriately to the alerts in the CIF report.

*Reply: Thank you for your constructive advice. Based on the single crystal tests and the related literatures, we believe that the given chiralities of the two compounds are correct. 1. exp_801: The method that asymmetric synthesis of chiral amines by highly diastereoselective 1,2-additions of organometallic reagents to N-tert-butanefulfinyl imines has been reported (Tetrahedron **1999**,55, 8883-8904). Such a method has been widely used in the Sadphos developed by our group, and similar ligands and their single crystals have also been obtained (Angew.Chem.Int.Ed.**2018**,57,10373–10377). In addition, the conditions introduced by borane are mild and will not react with other locations, the method is mature. 2. exp_2079: Since the chiral center is*

four-substituted carbon chiral, under the conversion conditions reported in the manuscript, it is generally believed that the chiral center does not participate in the reaction and will not change, so it is only the change of methoxy group and hydroxyl group relative to the target product. So we believe that the absolute configuration obtained is correct.

Response to Reviewer 4:

To gain insight into the reaction mechanism and the origin of stereo-selectivity, the authors have considered as an example the reaction described in Figure 6 of the main manuscript. By combining deuterium labelling with vibrational circular dichroism (VCD) spectroscopy, they have assigned the configuration of the resulting compound (8-D) to be SR. This in turn, confirmed a cis-oxypalladation operative mechanism. The assignment of the configuration of the 8-D compound is based on the observation that in the frequency interval between 1200 and 1400 1/cm, the VCD spectra computed for the SR and SS configurations of 8-D have bands of opposite signs. Since the spectrum computed for the SR configuration resembles the experiment better than the SS spectrum, the authors have concluded that the configuration of 8-D is SR. The procedure is correct and standard. Since the 8-D molecule is small I have performed test calculations for several 8-D-SR and 8-D-SS conformers. I can confirm that many of the conformers have in the SR and SS configurations VCD bands of opposite sign in the frequency interval between 1220 and 1330 1/cm. **Therefore, I believe that the assignment made by the authors is correct.** To be hundred percent sure, I ask the authors to provide some additional information and perform a few additional tests:

1. Show in the ESI also the comparison between the experimental and simulated IR spectra. To properly judge the relative sign of the SR and SS VCD bands and how similar they are with the experimental spectrum, the associated IR spectra need to be aligned with the experimental one.

Reply: Thank you for your constructive advice. We have performed IR spectrum tests

for 8-D, and DFT calculations for 8-D-SR and 8-D-SS, respectively (we performed both because deuterium may introduce minor differences into the IR spectra). Experimental and simulated spectra are provided in **Figure S1**. As a result, we reckon that the simulated IR spectra match well with the experimental one, which enhanced the confidence in our previous result.

Figure S1. Experimental and simulated IR spectra of 8-D-SR and 8-D-SS

2. Investigate how sensitive the prediction made with VCD is to a change of computational parameters. I am asking this because the simulated IR/VCD spectra are computed as a Boltzmann average of the spectra computed for the individual conformers, and the Boltzmann factors computed with DFT are inaccurate and very sensitive to the choice of computational parameters. Therefore, since not all conformers seem to exhibit VCD bands of opposite sign in the interval between 1200 and 1400 $1/\text{cm}$, it is important to verify how the SR and SS Boltzmann weighted VCD

spectra change when using Boltzmann factors predicted by different levels of theory. Since the experiment was performed in CDCl₃, I would suggest standard vacuum calculations (i.e., without dispersion correction and without specifying the chloroform as solvent) using the 6-31G (d, p) and def2-TZVP basis sets. This additional analysis is not expensive and should increase the level of confidence for the prediction made here with VCD.

Reply: Thank you for your constructive advice. As you suggested, we performed geometry optimization and frequency calculation on each conformation under different calculation levels.

*The Boltzmann distribution of all conformers calculated under different calculation levels is provided in **Table S1**. The VCD spectra are also provided in **Figure S2**. We observed that while there are variations in the Boltzmann distribution of different conformers and spectra, the overall trend remains consistent across various calculation levels. Therefore, we believe that this result lends support to our initial conclusion.*

Table S1. Boltzmann distribution profile of 8-D-S conformers under different computational parameters

structure index	Standard*	no-disp-scrf**	no-disp-scrf-dp***
0	4.80%	2.40%	2.51%
1	4.62%	4.93%	4.89%
2	6.96%	6.78%	6.90%
3	14.23%	14.60%	10.50%
4	11.03%	4.25%	4.69%
5	10.99%	4.25%	4.69%
6	4.74%	1.90%	1.96%
7	5.39%	6.17%	6.26%
8	0.77%	0.35%	0.26%
9	2.83%	3.99%	4.04%
10	0.44%	0.22%	0.20%

11	7.11%	12.98%	13.46%
12	8.69%	0.38%	0.43%
13	3.83%	7.70%	8.19%
14	3.63%	4.01%	4.23%
15	1.42%	3.90%	4.18%
16	1.55%	4.65%	4.73%
17	3.58%	10.66%	11.87%
18	0.52%	0.18%	0.18%
19	1.55%	4.69%	4.74%
20	0.71%	0.33%	0.33%
21	0.31%	0.34%	0.38%
22	0.31%	0.34%	0.38%

* standard parameter

** Standard parameter without dispersion correction and solvent model

*** standard parameter without dispersion correction and solvent model and change basis from 6-31G* to 6-31G** in the optimization process

Figure S2. VCD spectra of 8-D-SR and 8-D-SS under different computational parameters. (a) standard parameter; (b) standard parameter without dispersion correction and solvent model; (c) standard parameter without dispersion correction and solvent model and change basis from 6-31G* to 6-31G in the optimization process.**

3. Show in the ESI comparisons of the overlaid SR and SS VCD spectra of the

considered low-energy conformers.

Reply: Thank you for your suggestion. We have provided the overlaid and weighted SR and SS VCD spectra of all conformers in consideration in Figure S3.

Figure S3. Overlaid and weighted 8-D-SR (left) and 8-D-SS (right) VCD spectra of all conformers under standard calculation parameters. The red bold line is the calculated VCD spectra.

4. There seems to be an inconsistency regarding the compounds used in the VCD analysis. Section 6 in ESI mentions that the IR and VCD measurements were performed for compound 8-D, which is in line with what is written in the main manuscript. On the other hand, in Section 7 in ESI, it is stated that the IR and VCD calculations were performed for the 3a-D-SS and 3a-D-SR compounds. I assume this is a typo, however, the authors need to clarify/correct this point. Obviously, the calculations should be performed for the molecule on which the measurement was performed.

Reply: Thank you for pointing out the mistake. It is indeed a typo and we apologize for the mistake. All names regarding the compounds used in the VCD analysis are rechecked and corrected as 8-D (or 8-D-R, 8-D-S, 8-D-SR, 8-D-SS).

5. When describing the computational VCD procedure, the authors mention that conformational searches have been performed for both SR and SS configurations. This of course is not necessary, the SR and SS structure are obtained by simply exchanging a hydrogen atom with a deuterium atom, so one expects identical results from the two conformational searches. More worrying is, however, the fact that only 11 low-energy conformers have been found within an energy window of 2 kcal/mol. The

very basic conformational search I performed with RDKit has yielded 32 low-energy conformers for 8-D (which have different VCD spectra) within that same energetic window. Therefore, I am wondering whether the authors have considered all relevant conformers. What program has been used when performing the conformational search? The citations given by the authors (i.e., 22-24) don't seem to be correct as they do not describe a conformation search procedure/program.

*Reply: Thank you for your advice. Originally, we used a conformation search program developed by our group called **rpipmin**. However, this program is still under active development and has not been published yet. So, regarding your reasonable concern, we have turned to **CREST** program developed by Prof. Grimme for conformation search and used **isostat** program developed by Dr. Lu subsequently to cluster similar conformers. 23 S-conformers and 28 R-conformers are retrieved via this procedure. All results in the manuscript, SI, and this reply now are based on the new conformers. The detailed parameters used in **CREST** and **isostat** are provided in SI:*

VCD spectra of different conformers may differ considerably. To compare with the experimental spectrum, conformation searching and Boltzmann weighted spectra are essential. The workflow is provided below. First, systematic conformation searching of both configuration (S and R) of 8-D (8-D-S and 8-D-R) was performed with CREST program developed by Prof. Grimme.²² (detailed input: “crest xtbopt.xyz --T 4 --v3 --ewin 6 --rthr 0.35 --ethr 0.2 --chrg 0 --noreftopo --temp 298.15 --gfn 2 --shake 1 --hmass 1”) Then, isostat component of Molclus program developed by Tian Lu et al.²³ is used to remove the duplicated conformers (detailed input: “isostat crest_conformers.xyz -nt 4 -Nout 60 -Eout 5 -Edis 0.3 -Gdis 0.3 -T 298.15”). 23 and 28 low-energy conformations of 8-D-S and 8-D-R were achieved, respectively.

To be certain that we have considered enough conformers, the clustering threshold was tightened in isostat (modified input: “isostat crest_conformers.xyz -nt 4 -Nout 60 -Eout 5 -Edis 0.2 -Gdis 0.2 -T 298.15”), and 38 and 42 low-energy conformations of 8-D-S and 8-D-R were achieved. VCD calculation procedures were

applied to these conformers and the comparison between the 8-D-SS and 8-D-SR VCD spectra of different number of conformers are provided in **Figure S4**. From this, we may conclude that the conformers we have taken into consideration are basically sufficient and the spectra are reliable.

Figure S4. Comparison between VCD spectra calculated with different numbers of

conformers. (a) standard parameter without dispersion correction and solvent model, 23 conformers; (b) standard parameter without dispersion correction and solvent model, 38 conformers; (c) standard parameter, 23 conformers; (d) standard parameter, 38 conformers.

6. There are many errors associated with the citations in Section 7 in the ESI (i.e. on page S115). For example: 10(75), 11(76), 12–13(77, 78), etc.

Reply: *Thank you for pointing out these for us and we are sorry for the mistakes. We have revised and double-checked our citations in SI.*

REVIEWERS' COMMENTS

Reviewer #1 (Remarks to the Author):

The authors have properly answered all the comments I concerned in previous review. I have no further questions. This MS can be accepted without further changes.

Reviewer #2 (Remarks to the Author):

The manuscript under consideration details a Pd-catalyzed enantioselective alkoxyalkenylation of γ -hydroxyalkenes by ligand design and development. The method shows broad substrate generality, which is suitable for primary, secondary and tertiary hydroxyalkenes. Although related work has been previously reported using aryl halides, the current study with vinyl bromides is still considered to possess a degree of novelty and importance, especially considering the facile synthesis of enantio-enriched allyl-substituted tetrahydrofurans. Detailed mechanistic studies were conducted by the authors, which confirmed and verified the 'side arm effect' proposed by the authors. The content of the manuscript is well-presented and this work is recommended to be published in Nature Communications after providing comment on the following question or issue:

1. The authors states that the substituent group ortho to the side-arm (OBn) is crucial to the enantioselectivity of the reaction. Instead of using OMe and other alkoxy groups, did the authors synthesize ligands possessing methyl or ethyl groups ortho to the side arm (OBn), as these substituents are smaller than OiPr and OtBu in space, but more rigid than OMe group?
2. Impurities were observed in the HPLC traces of compound 8m. The enantio-enriched sample might be further purified, or the separation method could be optimized to avoid the overlap of the signals of impurity and the minor enantiomer (retention time at 13.585).

Reviewer #3 (Remarks to the Author):

This manuscript details a novel palladium-catalyzed enantioselective alkoxyalkenylation reaction of a range of primary, secondary, and tertiary γ -hydroxyalkenes with alkenyl halides and details two newly identified sulfinamide phosphine (Xu-Phos) ligands. The authors use VCD and DFT to verify the

cis-oxypalladation mechanism and located the stereo-determining TSs. In addition, the newly identified Xu-Phos ligands are structurally characterized using single-crystal X-ray diffraction techniques. While my expertise lies in crystallography, I believe this work does make a significant contribution to the field and should be published.

Reviewer #4 (Remarks to the Author):

Dear Editor,

In accordance with your request, I have conducted a thorough review of the VCD analysis presented in this manuscript, specifically focusing on its validation of the cis-oxypalladation operative mechanism. I have carefully examined the revised version of the NCOMMS-23-17932A manuscript and can confirm that the authors have effectively addressed all the concerns I raised in my initial evaluation. This solidifies my initial assessment that the VCD analysis conducted by the authors has indeed provided an accurate assignment.

Dear Referees,

*On behalf of my co-authors, I would like to express our sincere gratitude for your invaluable advice and suggestion on our manuscript titled " **Mechanistic study on the side arm effect in a Palladium/Xu-Phos-Catalyzed Enantioselective Alkoxyalkenylation of γ -Hydroxyalkenes** " (Manuscript ID: NCOMMS-23-17932). The portions of the manuscript and Supporting Information that have been modified are indicated by highlighting in "Yellow." Below, we have outlined our responses to the specific comments:*

Response to Reviewer 2:

Comment 1: The authors states that the substituent group ortho to the side-arm (OBn) is crucial to the enantioselectivity of the reaction. Instead of using OMe and other alkoxy groups, did the authors synthesize ligands possessing methyl or ethyl groups ortho to the side arm (OBn), as these substituents are smaller than OiPr and OtBu in space, but more rigid than OMe group?

Reply: Thank you for your advice. In response to your guidance, we would like to clarify that we had previously attempted to synthesize the ligands you recommended before submitting this paper's initial version. Regrettably, our efforts in this regard did not yield the desired products.

Comment 2: Impurities were observed in the HPLC traces of compound 8m. The enantio-enriched sample might be further purified, or the separation method could be optimized to avoid the overlap of the signals of impurity and the minor enantiomer (retention time at 13.585).

Reply: Thank you for your advice. The new HPLC graph has been updated in SI.

We want to take this opportunity to express our thanks to all referees for their careful reviewing, suggestions, which help us a lot of to improve the quality of this manuscript and make us gain insight of the research.

Department of Chemistry, Fudan University, 2005 Songhu Road, Shanghai,
200438, P. R. China

Best Regards,

Junliang Zhang

Fudan University,

Shanghai, P. R. China.

E-mail: junliangzhang@fudan.edu.cn